# Effect of Bt-Cry1Ab Maize Commercialization on Arthropod Community Biodiversity in Southwest China

**DOI:** 10.3390/insects16111132

**Published:** 2025-11-05

**Authors:** Limei He, Ling Wang, Yatao Zhou, Wenxian Wu, Shengbo Cong, Yanni Tan, Wei He, Gemei Liang, Kongming Wu

**Affiliations:** 1Institute of Urban Agriculture, Chengdu Agricultural Science and Technology Center, Chinese Academy of Agricultural Sciences, Chengdu 610299, China; 2State Key Laboratory for Biology of Plant Diseases and Insect Pests, Institute of Plant Protection, Chinese Academy of Agricultural Sciences, Beijing 100193, China; 3Key Laboratory of Integrated Pest Management on Crops in Central China, Ministry of Agriculture and Rural Affairs, Hubei Key Laboratory of Crop Disease, Insect Pests and Weeds Control, Institute of Plant Protection and Soil Science, Hubei Academy of Agricultural Sciences, Wuhan 430064, China

**Keywords:** transgenic insect-resistant maize, arthropod community, ecological index, similarity coefficient

## Abstract

**Simple Summary:**

Transgenic Bt maize cultivation has emerged as a critical pest management strategy against lepidopteran insects in China, but its ecological impact on arthropod biodiversity remains insufficiently characterized. Field assays demonstrate that Bt-Cry1Ab maize (DBN9936) cultivation effectively controls target pests without causing major alterations in overall community diversity, providing substantial empirical evidence to support its sustainable deployment in southern China’s agricultural landscapes.

**Abstract:**

Transgenic Bt maize commercialization has become a critical pest management strategy against lepidopteran insects in southwest China, but its ecological impact on arthropod biodiversity remains insufficiently characterized. This two-year field investigation (2023–2024) conducted in Bazhong City, Sichuan Province utilized systematic field monitoring to compare arthropod community dynamics between conventional maize and Bt-Cry1Ab maize (DBN9936) cultivation systems. This study documented 575,970 arthropod specimens representing 80 species/types across 45 families and 17 orders. Analysis of variance revealed significant differences (*p* < 0.05) between non-Bt and Bt maize in the abundance and species richness of target herbivorous pests, non-target herbivorous pests, and natural enemy insects. Field investigations revealed a notable absence of *Macrocentrus cingulum*, a key larval parasitoid of *Ostrinia furnacalis*, in Bt-maize plots compared to conventional counterparts. The populations of non-target herbivorous pests and natural enemies such as Aphididae, *Chrysoperla sinica*, *Frankliniella tenuicornis*, and *Orius sauteri* were higher in Bt maize fields than in non-Bt maize fields, while the populations of target herbivorous pests including *O. furnacalis* and *Mythimna loreyi* were lower than those in non-Bt maize fields. However, no significant differences (*p* > 0.05) were observed in arthropod abundance, species richness, or in a suite of ecological indices including the Simpson diversity index, Shannon–Wiener diversity index, Pielou evenness index, McIntosh diversity index, and community stability indices (*N*_n_/*N*_p_, *N*_d_/*N*_p_, and *S*_d_/*S*_p_). Redundancy analysis identified maize growth stages (6.75% variance explained) and interannual variations (2.44%) as principal drivers of arthropod community dynamics, with maize genotype contributing minimally (1.53%). These findings demonstrate that Bt-Cry1Ab maize (DBN9936) cultivation maintains functional arthropod community structure while effectively controlling target pests, providing substantial empirical evidence to support its sustainable deployment in southern China’s agricultural landscapes.

## 1. Introduction

The development and implementation of transgenic insect-resistant crops utilizing *Bacillus thuringiensis* (Bt)-derived insecticidal proteins have demonstrated multifaceted agricultural benefits. The insecticidal action of Bt-Cry toxins begins with their sequential binding to specific proteins in the insect larval midgut. This binding triggers the toxin’s oligomerization, enabling membrane insertion and pore formation, resulting in midgut cell death [1]. The destruction of gut wall cells allows intestinal bacteria to invade the hemocoel, triggering fatal septicemia in the insect. Beyond achieving the effective suppression of target pest populations, this biotechnology intervention significantly mitigates the incidence of bacterial rot disease while reducing reliance on chemical pesticides and associated environmental losses [2,3]. Substantial evidence from longitudinal studies confirms the critical role of Bt crops in enhancing agricultural sustainability through improved crop productivity and quality attributes, augmented farmer profitability, ecological conservation, and the facilitation of green agricultural transitions [2,3,4,5,6,7]. Following the pioneering adoption of Bt crops in the United States during 1996, transgenic crop cultivation has experienced exponential global expansion. Current agricultural systems incorporate 32 approved transgenic varieties spanning six major crops: maize, soybean, cotton, rapeseed, potato, and alfalfa. The latest agricultural census data reveal remarkable growth trajectories [8], with global transgenic crop coverage reaching 206.3 million hectares in 2023, representing a 121-fold increase since initial commercialization and occupying approximately 13.38% of global arable land [8]. This rapid adoption underscores the critical role of this technology in addressing contemporary agricultural challenges [4,5,9,10,11].

Maize is a critical global food security crop, ranking third in agricultural importance in China’s agricultural system [12]. Global maize production faces persistent biotic constraints from over 350 documented insect pest species, with China’s maize ecosystem harboring 230 confirmed pest taxa [13]. Lepidopteran, coleopteran, and hemipteran species constitute the predominant pest complexes, including but not limited to *Ostrinia furnacalis* Guenée, *Ostrinia nubilalis* Hübner, *Busseola fusca* Fuller, *Conogethes punctiferalis* Guenée, *Spodoptera exigua* Hübner, *Helicoverpa armigera* Hübner, *Spodoptera frugiperda* J.E. Smith, *Epilachna chrysomelina* Fabricius, *Monolepta signata* Olivier, and *Rhopalosiphum maidis* Fitch [13,14]. Notably, key pests including *O. furnacalis* and *H. armigera* inflict multidimensional damage through direct herbivory on foliar tissues and reproductive structures (tassels and ears), as well as the indirect facilitation of fungal pathogenesis (particularly *Fusarium verticillioides*-induced ear rot), resulting in substantial yield losses and quality degradation in commercial maize production [15,16,17]. Contemporary agricultural systems confront escalating phytosanitary challenges due to synergistic environmental pressures. Climate-mediated phenological shifts, transboundary pest invasions, and intensive monoculture practices have exacerbated the ecological dominance of destructive Lepidoptera species including *O. furnacalis* and the invasive *S. frugiperda*, establishing persistent threats to the security of maize cultivation across Asian agroecosystems. Current integrated pest management (IPM) strategies remain disproportionately dependent on synthetic insecticides. Nevertheless, decades of intensive pesticide application have engendered multifaceted agricultural challenges, including: (1) accelerated evolution of pest resistance mechanisms [5,18], (2) bioaccumulation of toxic residues in maize commodities exceeding food safety thresholds [5], (3) disruption of trophic interactions via non-target organism mortality [19,20], and (4) widespread ecological contamination affecting terrestrial and aquatic ecosystems [5,21,22]. These compounded effects collectively constrain the implementation of environmentally conscious agricultural practices and undermine the fundamental principles of sustainable maize production systems [5].

The cultivation of transgenic insect-resistant maize has emerged as a pivotal strategy in global maize production systems for managing lepidopteran pests, particularly *O. furnacalis* and *S. frugiperda* [5]. The global adoption rate of transgenic maize reached 34% of the total global maize cultivation area in 2023, reflecting its increasing agricultural significance [8]. In China, a milestone in agricultural biotechnology was achieved with the issuance of biosafety certificates for 20 transgenic maize varieties, including DBN9936, DBN9501, and Ruifeng 125. These certified varieties have been incorporated into structured pilot cultivation programs across China’s principal maize production zones, namely, southwest China, the Huang-Huai-Hai region, north China, and northeast China [23,24,25]. Among these genetically modified organisms, Bt maize DBN9936, which is engineered to express the Cry1Ab protein, represents China’s inaugural generation of commercially approved transgenic maize. Significantly, Bt cotton engineered to express the Cry1Ac protein was the pioneering Bt crop to be widely commercialized in China. This crop exerted a suppressive effect on *H. armigera* populations, reducing their incidence both in cotton fields and in non-Bt crops such as maize and peanuts [26]. Furthermore, Cry1Ab shares a high degree of structural similarity with Cry1Ac. Consequently, both toxins target the larvae of Lepidoptera insects [1]. Extensive field evaluations have demonstrated its efficacy against key lepidopteran pests, with mortality rates exceeding 90% for both *O. furnacalis* and *S. frugiperda* larva [27,28,29]. Regional cultivation trials of Bt maize DBN9936 commenced in 2023 across agriculturally significant provinces in China, including Inner Mongolia, Jilin, and Sichuan. However, comprehensive long-term monitoring remains crucial to evaluate the potential ecological consequences associated with the large-scale implementation of transgenic crops, particularly regarding non-target organism impacts and resistance evolution in pest populations.

Arthropods function as pivotal regulators of ecological equilibrium by mediating essential ecosystem services, including pollination networks, biological pest suppression, trophic-level food provisioning, and biogeochemical cycling via organic matter decomposition [21,22,30]. Within agricultural systems, these invertebrates constitute key functional components whose community integrity requires rigorous evaluation in transgenic crop risk assessments, essential for fulfilling biodiversity conservation commitments [31,32]. Over the past 25 years, hundreds of environmental impact studies on genetically modified crops conducted across Europe and the Americas have indicated that Bt corn exerts minimal, predominantly neutral effects on non-target invertebrate communities in cornfields [7,33,34]. A key finding was that robust and stable food webs persisted across diverse Bt maize fields, mirroring the patterns observed in conventional maize fields [35]. A significant lacuna exists, however, in the data from Asian agricultural systems. Empirical evidence from confined field trials in China has indicated that Bt maize cultivars exert no significant adverse impacts on arthropod diversity at experimental scales [19,36,37,38,39]. A recent study demonstrates that Bt maize has no adverse effects on predatory natural enemy diversity within the Yellow-Huai-Hai summer maize-growing region of China [40]. However, it remains unclear whether this finding holds under large-scale, real-world farming conditions, given the inherent variation in arthropod diversity across different agroecological regions. Sichuan Province presents an ideal study system for addressing this research priority, given its dual status as China’s principal maize production base (sustaining >1.8 million hectares of cultivation area with annual yields exceeding 10 million metric tons) and the designated pilot zone for Bt-Cry1Ab maize (DBN9936) cultivation. Through systematic field-based observational surveys across representative cultivation landscapes, this investigation evaluated the biodiversity implications of Bt-Cry1Ab maize (DBN9936) adoption at commercial production scales. The findings of this study provide data support for expanding DBN9936 (Bt-Cry1Ab) maize cultivation in China, as well as enrich the global repository of region-specific biosafety data on genetically modified crops.

## 2. Materials and Methods

### 2.1. Plants

The study utilized the Bt-Cry1Ab maize Zhongdan 808D (event DBN9936, Bt-maize), an early-adopted and widely promoted cultivar in Sichuan Province, China. Seed for this variety was provided by DBN Group (Beijing, China) in 2023 and by Beijing Abundance High-tech Seed Industry Co., Ltd. (Beijing, China) in 2024. For comparison, two non-Bt maize varieties (Dongdan 100 and Shuyu 336) conventionally planted by local farmers were included, sourced from Liaoning Dongya Seed Industry Co., Ltd. (Shenyang, China) and Sichuan Shuyu Technology Agriculture Development Co., Ltd. (Chengdu, China), respectively.

### 2.2. Sampling Site

The Chinese government initiated pilot programs for Bt-maize commercialization in Xueshan Town and Shangbamiao Town of Enyang District, Bazhong City, Sichuan Province, in 2023 and 2024, respectively. Field investigations were carried out in 2023 across two villages in Xueshan Town, Enyang District, Bazhong City, Sichuan Province, China (Figure 1). Specifically, 24 Bt maize fields were randomly selected from Sanyan Village, which were fully dedicated to Bt maize cultivation. For comparison, 12 non-Bt maize fields were selected from Qinglong Village (the nearest village with exclusive non-Bt cultivation), where the primary cultivar was the non-Bt variety Shuyu 336. In 2024, both non-Bt maize (Dongdan 100, local cultivated varieties) and Bt maize fields were randomly selected from Yuhuangguan Village, Shangbamiao Town, Enyang District, Bazhong City, Sichuan Province, China (Figure 1). All maize varieties were manually sown on March 15 of each year, and the area of each plot was 50 m^2^ (6–7 rows, 7–8 m per row). The plant spacing and row spacing were 0.6 m and 0.3 m, respectively. Urea (20 kg/667 m^2^, N ≥ 46.2%, Sichuan Lutianhua Co., Ltd., Luzhou, China) and compound fertilizer (15 kg/667 m^2^, C-N-P ≥ 45%, Stanley Chemical Fertilizer Co., Ltd., Linyi, China) were applied at the late maize seedling stage. In the non-Bt maize fields, chlorantraniliprole (10 mL/667 m^2^, 200 g/L, Anyang Ruipu Agrochemical Co., Ltd., Anyang, China) and alphacypermethrin (25 mL/667 m^2^, 230 g/L; Shandong Wanhao Chemical Co., Ltd., Dongying, China) were applied for lepidopteran pest control during the seedling stage (V8–V10), while chemical insecticides were not applied to the Bt-Cry1Ab maize field. During the tasseling and silking stages (VT–R1), both Bt maize and non-Bt maize fields were treated with imidachloprid (5 g/667 m^2^, 700 g/kg; Shenzhen Nuopuxin Crop Science Co., Ltd., Shenzhen, China) and thiamethoxam (5 g/667 m^2^, 750 g/kg; Shandong Huimin Zhonglian Biotechnology Co., Ltd., Jinan, China) for aphid control.

### 2.3. Arthropod Community Species and Population Data Collection

Thirty days after sowing, the species composition and population size of the arthropod community were surveyed and recorded using a W-shaped five-spot sampling method with 20 plants randomly sampled for each location (for a total of 100 plants per sampling event) [6]. The numbers of visible arthropods on leaves, ears, husks, stalks, sheaths, and tassels were noted, and the total numbers per plot were counted. To maintain practical feasibility and data consistency, species with stable morphological characteristics (e.g., *O. furnacalis*, *H. armigera*) were identified to the species level, whereas taxa like aphids, which lack reliable field characteristics, were recorded at higher taxonomic levels (genus or family). Field surveys were conducted every 15 days, with a total of seven field surveys conducted per year by the end of the experimental period. The maize growth stages during the first through seventh surveys consisted of the four-to-five-leaf seedling stage (V4–V5) the six-to-eight-leaf seedling stage (V6–V8), the nine-to-twelve-leaf seedling stage (V9–V12), the fourteen-leaf seedling stage or tasseling stage (V14–VT), the silking stage or blister stage (R1–R2), the milk stage or dough stage (R3–R4), and the dent stage or physiological maturity stage (R5–R6). Arthropods were classified as target herbivorous insects, non-target herbivorous insects, natural enemy insects, and neutral insects based on the trophic level of the food chain and the target of Bt maize [19].

### 2.4. Statistical Analysis

The Simpson diversity index (*D*), Shannon–Wiener diversity index (*H*), and McIntosh diversity index (*D*_Mc_) were employed to analyze the biodiversity of the arthropod community in Bt and non-Bt maize fields. The Pielou evenness index (*J*) was used to analyze the evenness of the arthropod community, and the Berger–Parker dominance index was utilized to analyze the species dominance in the arthropod community (*I*). The four ecological indices employed provide complementary insights into community structure. The Simpson diversity index emphasizes dominant species while being less sensitive to rare ones. In contrast, the Shannon–Weiner diversity index integrates measures of both species richness and evenness. The Pielou evenness index offers a pure assessment of evenness by controlling for richness. Finally, the McIntosh diversity index conceptualizes a community as a point in multidimensional space, simultaneously influenced by richness and evenness, thereby capturing fundamental differences among communities. Reporting all four indices concurrently enhances the robustness and comprehensiveness of our conclusions. These ecological indices were calculated using the following formulas [41,42,43]: $D=1-\sum_{i=1}^{s}\left[\frac{{n_{i}(n}_{i}-1)}{N\left(N-1\right)}\right]$, $H=-\sum_{i=1}^{s}P_{i}\ln\left(P_{i}\right)$, $J=\frac{H}{H_{max}}=\frac{H}{\ln s}$, $D_{Mc}=\frac{\left(N-\sqrt{\sum_{i=1}^{s}n_{i}}\right)}{\left(N-\sqrt{N}\right)}$, $I=\frac{n_{i}}{N}$, where *D* is the Simpson diversity index; *n_i_* is the number of individuals in the *i*th species/type; *N* is the total number of individuals; *s* is the total number of species/type; *H* is the Shannon–Wiener diversity index; *P_i_* is the proportion of individuals belonging to the *i*th species/type in the total number of individuals; *J* is the Pielou evenness index; *D*_Mc_ is the McIntosh diversity index; and *I* is the Berger–Parker dominance index, where *I* ≥ 0.10 indicates a dominant species, 0.01 ≤ *I* < 0.10 indicates a common species, *I* < 0.01 indicates a rare species.

The Sørensen index was employed to analyze the similarity of the arthropod community between Bt and non-Bt maize fields. The calculation formula is $C=\frac{2W}{\left(a+b\right)}$, where *C* is the Sørensen index, *W* is the number of species found in both Bt and non-Bt maize fields, *a* and *b* are the total number of species found in Bt and non-Bt maize fields, respectively. A Sørensen index value of 0 < *C* < 0.25 indicates significant differences, 0.25 ≤ *C* < 0.5 indicates moderate differences, 0.5 ≤ *C* < 0.75 indicates moderate similarities, and 0.75 ≤ *C* ≤ 1 indicates high similarities [44].

Gao et al. [45] utilized two ratios to evaluate community stability: *S*_s_/*S*_i_ and *S*_n_/*S*_p_, where *S*_s_ represents the number of species, *S*_i_ represents the number of individuals, *S*_n_ represents the number of natural enemy species, and *S*_p_ represents the number of phytophagous insect species. A high *S*_s_/*S*_i_ ratio suggests that a community is characterized by high biodiversity and low population densities, leading to strong quantitative constraints among species. Similarly, a high *S*_n_/*S*_p_ ratio implies a greater regulatory influence from natural enemies, resulting in a more complex food web with intensified trophic interactions, whereby stability is enhanced. In this study, stability indices of the arthropod community (*N*_n_/*N*_p_, *N*_d_/*N*_p_, *S*_n_/*S*_p_, and *S*_d_/*S*_p_) were utilized to analyze the regulatory effects of natural enemies and neutral insects on herbivorous insects. In the stability indices, *N*_n_, *N*_p_, and *N*_d_ represent the individual numbers of natural enemy insects, herbivorous insects, and neutral insects, respectively, while *S*_n_, *S*_p_, and *S*_d_ represent the species numbers of natural enemy insects, herbivorous insects, and neutral insects, respectively [19,46].

DPS software version 9.01 was employed to investigate the diversity indices of the arthropod community in Bt and non-Bt maize fields [47]. The effects of maize type (Bt vs. non-Bt), growth stage, and year on the abundance, diversity, and stability indices of the arthropod community were analyzed using the general linear model (GLM) followed by Duncan’s new multiple range test (MRT) or simple effect analysis (SEA), with data first log-transformed to meet the assumptions of normality and heteroscedasticity. GLM was conducted using SPSS version 26.0 (IBM, Armonk, NY, USA). Redundancy analysis (RDA) was performed to discern the possible relationships between the ecological indices of the arthropod community and the maize type (Bt vs. non-Bt), growth stage, and year. RDA was conducted using Canoco version 5.0 [48,49].

## 3. Results

### 3.1. Arthropod Community Structure in Bt and Non-Bt Maize Fields

From 2023 to 2024, a total of 575,970 arthropods of 80 species/types, belonging to 45 families and 17 orders, were observed in Bt and non-Bt maize fields (Appendix A). In 2023, 140,365 individuals (63 species/types) were observed in non-Bt maize fields, and 283,414 individuals (63 species/types) were recorded in Bt maize fields. In 2024, 77,106 individuals (38 species/types) were observed in non-Bt maize fields, and 75,085 individuals (38 species/types) were recorded in Bt maize fields. The observations revealed 15 species/types of target herbivorous insects (including *O. furnacalis*, *Mythimna loreyi* Duponchel, *Peridroma saucia* Hübner, and *H. armigera*), 35 species/types of non-target herbivorous insects (including Aphididae, *Tetranychus cinnabarinus* Biosduval, and *Frankliniella tenuicornis* Uzel), 24 species/types of natural enemy insects (including *Chrysoperla sinica* Tjeder, *Orius sauteri* Poppius, *Harmonia axyridis* Palla, *Propylaea japonica* Thunberg, and *Lysaphidus* sp.), and 6 species/types of neutral insects (including Formicidae and *Pieris rapae* L.). The occurrences of different arthropods were similar in Bt and non-Bt maize fields over the two years, and the dominant species/types consisted of Aphididae and *T. cinnabarinus* (Appendix A).

The number of non-target herbivorous insect individuals or species (Figure 2B and Figure 3B; Appendix A), neutral insect individuals or species (Figure 2D and Figure 3D; Appendix A) and arthropod community individuals or species (Figure 2E and Figure 3E; Appendix A) demonstrated significant differences among the growth stages of maize and the year (*p* < 0.05; Appendix A). The number of target herbivorous insect individuals or species (Figure 2A and Figure 3A; Appendix A) and natural enemy insect individuals or species (Figure 2C and Figure 3C; Appendix A) was not significantly different over the two years, but varied significantly in maize growth stages (*p* < 0.05; Appendix A).

The number of target herbivores (Figure 2A; Appendix A), non-target herbivores (Figure 2B; Appendix A), natural enemy (Figure 2C; Appendix A) and neutral insect individuals (Figure 2D; Appendix A) showed significant differences between Bt and non-Bt maize fields (*p* < 0.05; Appendix A), but the number of arthropods exhibited no significant differences between Bt and non-Bt maize fields (*p* > 0.05; Figure 2E; Appendix A). In Bt maize fields, the abundances of target herbivores, non-target herbivores, and neutral insects decreased by 40.69%, 8.04%, and 50.54%, respectively, while the natural enemy abundance increased by 86.38%, compared to non-Bt maize fields.

The abundance of target herbivores, non-target herbivores, natural enemies, neutral insects and arthropods differed significantly among year × growth stage interactions (*p* < 0.05; Appendix A). Significant variation in the number of target herbivores, non-target herbivores, natural enemies, neutral insects, and arthropods across growth stages was consistently observed in both 2023 (target herbivores: *F*_6, 280_ = 12.060, *p* < 0.001; non-target herbivores: *F*_6, 280_ = 70.665, *p* < 0.001; natural enemies: *F*_6, 280_ = 47.802, *p* < 0.001; neutral insects: *F*_6, 280_ = 61.961, *p* < 0.001; arthropods: *F*_6, 280_ = 106.084, *p* < 0.001) and 2024 (target herbivores: *F*_6, 280_ = 8.566, *p* < 0.001; non-target herbivores: *F*_6, 280_ = 31.628, *p* < 0.001; natural enemies: *F*_6, 280_ = 22.943, *p* < 0.001; neutral insects: *F*_6, 280_ = 7.258, *p* < 0.001; arthropods: *F*_6, 280_ = 11.610, *p* < 0.001) (Appendix A). Target herbivores’ abundance demonstrated significantly differences between 2023 and 2024 during R1–R2 (*F*_1, 280_ = 9.799, *p* = 0.002), R3–R4 (*F*_1, 280_ = 4.770, *p* = 0.030), and R5–R6 (*F*_1, 280_ = 9.032, *p* = 0.003) (Appendix A). Non-target herbivores’ abundance demonstrated significantly differences between 2023 and 2024 during V6–V8 (*F*_1, 280_ = 12.378, *p* = 0.001), R1–R2 (*F*_1, 280_ = 13.208, *p* < 0.001), and R3–R4 (*F*_1, 280_ = 26.525, *p* < 0.001) (Appendix A). The abundance of natural enemies demonstrated significantly differences between 2023 and 2024 during V6–V8 (*F*_1, 280_ = 31.140, *p* < 0.001) and R1–R2 (*F*_1, 280_ = 6.002, *p* = 0.015) (Appendix A). The abundance of neutral insects demonstrated significantly differences between 2023 and 2024 during V14–VT (*F*_1, 280_ = 28.110, *p* < 0.001; Appendix A). Arthropods abundance demonstrated significantly differences between 2023 and 2024 during V6–V8 (*F*_1, 280_ = 61.530, *p* < 0.001), V9–V11 (*F*_1, 280_ = 23.830, *p* < 0.001), V14–VT (*F*_1, 280_ = 9.604, *p* = 0.002), and R1–R2 (*F*_1, 280_ = 6.830, *p* = 0.009) (Appendix A).

The abundance of target herbivores, non-target herbivores, neutral insects and arthropods differed significantly among year × maize type interactions (*p* < 0.05; Appendix A). The abundance of target herbivores, non-target herbivores, and arthropods in 2023 varied significantly between Bt and non-Bt maize fields (target herbivores: *F*_1, 290_ = 50.558, *p* < 0.001; non-target herbivores: *F*_1, 290_ = 13.847, *p* < 0.001; arthropods: *F*_1, 290_ = 5.043, *p* = 0.022), whereas it remained relatively constant in 2024 (target herbivores: *F*_1, 290_ = 0.062, *p* = 0.804; non-target herbivores: *F*_1, 290_ = 0.012, *p* = 0.911; arthropods: *F*_1, 290_ = 0.022, *p* = 0.882) (Appendix A). The abundance of neutral insects in Bt maize fields varied significantly over the two years (*F*_1, 290_ = 9.875, *p* = 0.002), whereas it remained relatively constant in non-Bt maize fields (*F*_1, 290_ = 0.490, *p* = 0.484) (Appendix A).

The abundance of target herbivores and natural enemies differed significantly among growth stage × maize type interactions (*p* < 0.05; Appendix A). Furthermore, significant variation in target herbivores and natural enemies’ abundance across growth stages was consistently observed in both Bt (target herbivores: *F*_6, 280_ = 9.961, *p* < 0.001; natural enemies: *F*_6, 280_ = 51.405, *p* < 0.001) and non-Bt maize fields (target herbivores: *F*_6, 280_ = 15.068, *p* < 0.001; natural enemies: *F*_6, 280_ = 15.756, *p* < 0.001) (Appendix A). The abundance of target herbivores demonstrated significant differences between non-Bt and Bt maize fields during V6–V8 (*F*_1, 280_ = 4.339, *p* = 0.038), V9–V11 (*F*_1, 280_ = 45.729, *p* < 0.001), V14–VT (*F*_1, 280_ = 6.180, *p* = 0.014), R3–R4 (*F*_1, 280_ = 21.559, *p* < 0.001), and R5–R6 (*F*_1, 280_ = 16.606, *p* < 0.001) (Appendix A). The abundance of natural enemies demonstrated significant differences between non-Bt and Bt maize fields during V6–V8 (*F*_1, 280_ = 5.520, *p* = 0.019), R3–R4 (*F*_1, 280_ = 9.550, *p* = 0.002), and R5–R6 (*F*_1, 280_ = 19.314, *p* < 0.001) (Appendix A). Furthermore, during the R1–R2, the non-target herbivores’ and arthropod abundances in non-Bt maize fields were 3.4 and 3.2 times greater, respectively, than those in Bt maize fields. In contrast, during the R5–R6, the abundance of non-target herbivores, natural enemy and arthropod in Bt maize fields were 5.6, 5.2 and 4.1 times greater, respectively, than those in non-Bt maize fields.

The number of neutral insect species (Figure 3D; Appendix A) and arthropod community species (Figure 3E; Appendix A) demonstrated no significant differences between Bt and non-Bt maize fields (*p* > 0.05; Appendix A). The number of target herbivorous insect species (*p* < 0.05; Figure 3A; Appendix A; Appendix A), non-target herbivorous insect species (*p* < 0.05; Figure 3B; Appendix A), and natural enemy insect species (*p* < 0.05; Figure 3C; Appendix A) differed significantly between Bt and non-Bt maize fields. Especially, In Bt maize fields, the number of target herbivorous insect species decreased by 62.72%, while the numbers of non-target herbivorous insects and natural enemy species increased by 32.32% and 13.29%, respectively, compared to non-Bt maize fields.

The number of species for target herbivores, non-target herbivores, natural enemies and arthropods differed significantly among year × growth stage interactions (*p* < 0.05; Appendix A). Furthermore, significant variation in the number of target herbivorous insect species, non-target herbivore herbivorous insect species across growth stages was consistently observed in both 2023 (target herbivores: *F*_6, 280_ = 29.014, *p* < 0.001; non-target herbivores: *F*_6, 280_ = 41.655, *p* < 0.001; natural enemies: *F*_6, 280_ = 35.224, *p* < 0.001; arthropods: *F*_6, 280_ = 106.084, *p* < 0.001) and 2024 (target herbivores: *F*_6, 280_ = 7.956, *p* < 0.001; non-target herbivores: *F*_6, 280_ = 8.834, *p* < 0.001; natural enemies: *F*_6, 280_ = 21.725, *p* < 0.001; arthropods: *F*_6, 280_ = 42.515, *p* < 0.001) (Appendix A). The number of target herbivore herbivorous insect species demonstrated significant differences between 2023 and 2024 during R1–R2 (*F*_1, 280_ = 4.157, *p* = 0.042) (Appendix A). The number of non-target herbivore herbivorous insect species demonstrated significant differences between 2023 and 2024 during V6–V8 (*F*_1, 280_ = 12.910, *p* < 0.001), V9–V11 (*F*_1, 280_ = 15.853, *p* < 0.001), and V14–VT (*F*_1, 280_ = 7.235, *p* = 0.008) (Appendix A). The number of natural enemy insect species demonstrated significant differences between 2023 and 2024 during V6–V8 (*F*_1, 280_ = 37.720, *p* < 0.001) and R3–R4 (*F*_1, 280_ = 4.190, *p* = 0.042) (Appendix A). The number of arthropods species demonstrated significant differences between 2023 and 2024 during V6–V8 (*F*_1, 280_ = 61.530, *p* < 0.001), V9–V11 (*F*_1, 280_ = 23.830, *p* < 0.001), V14–VT (*F*_1, 280_ = 9.604, *p* = 0.002), and R1–R2 (*F*_1, 280_ = 6.830, *p* = 0.009) (Appendix A). The number of species for neutral insects differed significantly among year × maize type (*p* < 0.05; Appendix A). Significant variation in the number of species for neutral insects between non-Bt and Bt maize fields was observed in both 2023 (*F*_1, 290_ = 5.484, *p* = 0.020) and 2024 (*F*_1, 290_ = 4.165, *p* = 0.042) (Appendix A).

During the same growth stage of maize, the proportion of individuals or species of target herbivorous insects, non-target herbivorous insects, natural enemy insects and neutral insects was roughly the same in Bt and non-Bt maize fields (Figure 4A,B). The highest proportion of individuals or species was recorded for herbivorous (target and non-target) insects, followed by natural enemy insects, while neutral insects had the lowest proportion.

### 3.2. Arthropod Community Diversity in Bt and Non-Bt Maize Fields

The Shannon–Wiener diversity index was not significantly different between Bt and non-Bt maize fields (*p* > 0.05) but varied significantly with the growth stage of maize and the year (*p* < 0.05; Figure 5A; Appendix A). The Simpson diversity index, Pielou evenness index, and McIntosh diversity index showed no significant differences between maize type (Bt vs. non-Bt) and year (*p >* 0.05) but varied significantly with the growth stage of maize (*p* < 0.05; Figure 5B–D; Appendix A).

### 3.3. Similarity of the Arthropod Community in Bt and Non-Bt Maize Fields

Similarity analysis indicated that the composition of the arthropod communities in Bt and non-Bt maize fields showed moderate differences or similarities during the nutritional growth stage of maize, with Sørensen indices ranging from 0.27 to 0.75 (Table 1). During the reproductive growth stage of maize, the arthropod community of Bt maize fields was highly similar to that of non-Bt maize fields, with Sørensen indices ranging from 0.64 to 0.85 (Table 1). In 2023, the arthropod community, target herbivorous group, non-target herbivorous group, and natural enemy group between Bt and non-Bt maize fields exhibited the highest similarity at stages V14–VT, R1–R2, V14–VT, and R5–R6, with Sørensen indices of 0.85, 0.89, 0.88, and 0.93, respectively. In 2024, the arthropod community, target herbivorous groups, and natural enemy groups showed the highest similarity at R3–R4, with Sørensen indices of 0.77, 0.80, and 0.84, respectively, while the non-target herbivorous groups demonstrated the highest similarity at V6–V8 (0.67).

### 3.4. Stability of the Arthropod Community in Bt and Non-Bt Maize Fields

There was little difference in the stability of arthropod communities between Bt and non-Bt maize fields (Appendix A). The *S*_n_/*S*_p_ stability index exhibited significant differences between Bt and non-Bt maize fields (*p* < 0.05), while the differences were not significant among growth stage of maize and year (*p* > 0.05; Figure 5H; Appendix A). The *N*_d_/*N*_p_ and *S*_d_/*S*_p_ stability indices exhibited no significant differences among the maize type (Bt vs. non-Bt), growth stage, and year (*p* > 0.05; Figure 5E,G; Appendix A). The *N*_n_/*N*_p_ stability index showed no significant differences between maize type (Bt vs. non-Bt) (*p* > 0.05) but varied significantly with the growth stage of maize and the year (*p* < 0.05; Figure 5F; Appendix A). Overall, the arthropod community stability indices (*N*_n_/*N*_p_, *N*_d_/*N*_p_, *S*_n_/*S*_p_, and *S*_d_/*S*_p_) in both Bt and non-Bt maize fields remained relatively stable.

### 3.5. Correlation Analysis Between Environmental Factors and Arthropod Diversity or Stability

Maize type (Bt vs. non-Bt), growth stage, and year in total explained 10.70% of the variance in the abundance data (*p* < 0.05; Appendix A), with 6.75% and 2.44% of the variance explained by the first and second axes, respectively (Figure 6A; Appendix A). The maize type (Bt vs. non-Bt), growth stage, and year made significant contributions to the variance at 15.00%, 62.80%, and 22.2%, respectively (*p* < 0.05; Figure 6A; Appendix A). For simplicity, Figure 6A only presents species/types for the top 13 highest species weights. The abundances of Aphididae, Araneae, Formicidae, *C. sinica*, *F. tenuicornis*, *O. sauteri*, and *Telenomus remus* were positively correlated with maize type (Bt vs. non-Bt), while the abundances of *O. furnacalis*, *M. loreyi*, *T. cinnabarinus*, *Nezara viridula*, *H. axyridis*, and *P. japonica* were negatively correlated with maize type (Bt vs. non-Bt) (Figure 6A).

Maize type (Bt vs. non-Bt), growth stage, and year explained 28.10% of the variance of ecological indices in total (*p* < 0.05; Appendix A), with 26.95% and 0.73% of the variance explained by the first and second axes, respectively (*p* < 0.05; Figure 6B; Appendix A). The variance contributions of maize type (Bt vs. non-Bt) and growth stage were significant at 4.50% and 92.7%, respectively (*p* < 0.05; Figure 6B; Appendix A). The number of individuals and species were positively correlated with maize type (Bt vs. non-Bt) and growth stage. However, the Shannon–Wiener diversity index, Simpson diversity index, McIntosh diversity index and Pielou evenness index were negatively correlated with maize type (Bt vs. non-Bt) and growth stage (Figure 6B).

Maize type (Bt vs. non-Bt), growth stage, and year explained 6.40% of the variance of stability indices in total (*p* < 0.05; Appendix A), with 5.44% and 0.67% of the variance explained by the first and second axes, respectively (Figure 6C; Appendix A). The variance contribution of maize type (Bt vs. non-Bt) (10.50%) was not significant (*p* > 0.05), while the year and growth stage of maize made significant variance contributions of 16.30% and 73.20%, respectively (*p* < 0.05; Figure 6C; Appendix A).

In summary, the abundance, diversity and stability indices of the arthropod community in the field were highly correlated with the growth stage of maize and the year but less correlated with maize type (Bt vs. non-Bt).

## 4. Discussion

While current evidence from Europe and North America suggests that the widespread adoption of Bt maize maintains arthropod biodiversity in both field and soil ecosystems [34,35,50], the potential long-term risks of its large-scale cultivation in China remain poorly understood [11,31,51]. Preliminary studies on small-scale experimental plantings of Bt maize cultivars (e.g., DBN318) in China have shown comparable arthropod community profiles to conventional maize [19,37,38]. However, China’s vast territorial expanse (9.6 million km^2^) encompasses diverse ecological zones with distinct topographies: southern regions (22–34° N), including southwest and south China, feature complex landscapes of mountains, hills, and basins, whereas northern areas (34–53° N), such as the North China Plain and northeast China, are dominated by plains and plateaus. Benefiting from tropical and subtropical monsoon climates, arthropod populations in southern regions thrive with extended activity periods and ample nourishment, thereby sustaining significant speciation and pronounced biodiversity [52,53]. Conversely, the temperate monsoon and continental climates of northern areas permit only those species endowed with specialized adaptive traits to endure, which consequently limits total species richness [52,53]. This geoclimatic heterogeneity raises unanswered questions about the ecological impacts of Bt maize cultivation across different bioregions. To address this knowledge gap, this study comprehensively evaluated the effects of Bt maize and non-Bt maize production patterns on arthropod community biodiversity in Sichuan Province, a representative southern maize production region characterized by a subtropical monsoon climate and a mosaic agricultural landscapes.

The present study revealed that the arthropod community in maize fields of Enyang District, Bazhong City, Sichuan Province, comprised 80 species/types across 17 orders and 45 families. Comparative analysis between non-Bt maize and Bt maize DBN9936 fields showed that herbivorous groups (including both target and non-target pests) consistently exhibited the highest population density and species richness within the arthropod communities, followed by natural enemy groups, while neutral groups remained relatively scarce. Notably, the community composition and structure showed remarkable similarities between non-Bt and Bt maize ecosystems. Statistical comparisons revealed no significant differences in key ecological indices between the two systems, including the Simpson diversity index, Shannon–Wiener diversity index, Pielou evenness index, McIntosh richness index, and community stability indices (*N*_n_/*N*_p_, *N*_d_/*N*_p_, and *S*_d_/*S*_p_). The results support the conclusion that Bt maize cultivation poses minimal risk to the structure and function of the field arthropod community, as evidenced by non-significant changes in both its overall ecological indices and the trophic interactions involving natural enemies and neutral insects. These findings align with previous studies on Bt maize varieties IE09S034, HGK60, and BT799 [19,36,37] and corroborate Priestly and Brownbridge’s conclusion that environmental conditions and agricultural management practices exert greater influence on arthropod communities than crop genotype [54]. RDA identified the maize growth stage and interannual variation as primary determinants of arthropod community dynamics, accounting for 84% of observed variability, while maize type (Bt vs. non-Bt) contributed minimally. This comprehensive assessment confirmed that Bt-Cry1Ab maize DBN9936 cultivation maintained arthropod biodiversity and showed ecological equivalence to conventional maize, which is consistent with previously reported field evaluation results for Bt maize DBN9936 and Ruifeng 125 in the Huanghuaihai Plain of China [40]. Our preliminary research showed that planting Bt-Cry1Ab maize DBN9936 efficiently reduced the population of *O. furnacalis* larvae and the percentage of damaged maize plants in Sichuan Province, China [6]. The current results provide robust empirical support for the large-scale commercialization of Bt-Cry1Ab maize DBN9936 in southwest China, the development of sustainable pest management strategies, and the refinement of environmental risk assessment frameworks for transgenic crops.

As the foundation for population growth, the larval stage involves direct tissue consumption, where nutrition critically determines future pupal and adult fitness [55,56,57]. Adult feeding on resources like nectar, however, fuels reproduction and migration, thereby setting the scale and range of the next generation [58,59,60]. Therefore, disrupting both life stages is essential for sustainable IPM. It is noteworthy that Bt insect-resistant crops primarily target the larval stage [1]. Since 1996, transgenic crops expressing Bt insecticidal proteins have achieved global adoption, demonstrating remarkable efficacy in controlling key lepidopteran and coleopteran agricultural pests [9,26,61]. However, this technological advance has sparked persistent scientific debate regarding the potential ecological consequences of transgenic crops on arthropod biodiversity in agricultural ecosystems. Current empirical evidence presents a dichotomy: comparative analyses reveal structural similarities between Bt-maize and non-Bt maize agroecosystems, particularly in their maintenance of complex trophic networks [35]. In contrast, multi-trophic studies suggest that transgenic crops may indirectly influence non-target insect populations through cascading effects on plant–herbivore–natural enemy interactions [62,63]. Some scholars also believe that genetically modified crops may further reduce biodiversity by strengthening agricultural intensification [64]. Our findings showed that Bt maize fields had 62.72% fewer target herbivorous insect species than non-Bt maize fields. In contrast, the number of non-target herbivorous insect and natural enemy species increased by 32.32% and 13.29%, respectively. In 2023, our field investigations revealed a conspicuous absence of *Macrocentrus cingulum*, a key larval parasitoid of *O. furnacalis*, in Bt-maize (DBN9936) plots compared to non-Bt maize plots. This phenomenon likely stems from the rapid Bt protein-induced mortality of neonate *O. furnacalis* larvae, thereby eliminating suitable hosts for parasitoid development. RDA showed non-significant associations between maize type (Bt vs. non-Bt) and community-level metrics, namely species richness, diversity indices, and stability coefficients. However, populations of non-target herbivorous pests (e.g., Aphididae) and natural enemies (e.g., *C. sinica*, *F. tenuicornis*, *O. sauteri*) were significantly elevated in Bt maize fields, whereas target herbivores (e.g., *O. furnacalis*, *Mythimna lorei*) exhibited marked suppression. These findings indicate that Bt-maize cultivation triggers compensatory ecological mechanisms by: (1) direct suppression of target Lepidoptera reducing insecticide applications, (2) subsequent ecological release promoting secondary pest (e.g., aphids, *N. viridula*) proliferation, and (3) bottom-up effects enhancing natural enemy recruitment.

Studies from the US, Brazil, China, and other countries find that Bt maize’s effects on non-target invertebrate communities in maize fields are typically minor and neutral, especially compared to those of broad-spectrum pyrethroid insecticides [7,33,37,39]. Notably, current risk assessment paradigms predominantly focus on arthropod community composition [36,37,40], while largely neglecting tri-trophic interaction dynamics within plant–herbivore–natural enemy. To advance the mechanistic understanding of transgenic crop impacts, future research should prioritize: (1) implementing multi-trophic level analyses of energy flow, (2) developing quantitative food web models, (3) quantifying long-term selection pressures on non-target species, and (4) establishing bio-indicator systems for ecological risk assessment. This paradigm shift from community-level descriptors to functional interaction networks will facilitate more predictive assessments of agricultural biotechnology’s ecological consequences.

The cultivation of Bt genetically modified insect-resistant maize represents an environmentally sustainable agricultural practice [2,3,4,6,50]. However, prolonged and widespread adoption of Bt crops has progressively induced resistance development in target pests. The emergence of practical resistance in 11 target pest species across seven countries poses a serious threat to the continued effectiveness and operational lifespan of Bt crops [65,66]. Furthermore, Chinese research indicates that while Bt cotton has successfully suppressed populations of target pests such as *H. armigera*, it has concurrently promoted secondary pests like the Miridae from minor to major status in cotton agroecosystems [67]. Additional studies suggest that genetically modified crops may contribute to an overall increase in pesticide consumption while perpetuating chemical-intensive monocultural farming systems [68]. This phenomenon may occur as Bt crop cultivation leads to the emergence of secondary non-target pests or weeds as primary concerns, consequently driving increased chemical interventions against these non-target species [67]. Our results also showed a higher population density of non-target insects in Bt maize fields compared to non-Bt maize fields. The abundance of target herbivores, non-target herbivores, natural enemies, neutral insects and arthropods varied significantly among the interaction effects of year × growth stage, year × maize type, or growth stage × maize type. A plausible explanation for the observed patterns involves the application of the lepidopteran-specific insecticide chlorantraniliprole in non-Bt maize fields, which was absent in Bt plots. This insecticide may have broadly suppressed a range of non-target insects beyond the intended lepidopteran pests. In addition, geographical variation across sampling sites and differences in replication numbers may have contributed to the ecological differences detected between the two cropping systems. Therefore, strategic promotion and cultivation of Bt crops should incorporate ongoing surveillance of resistance evolution in target organisms, alongside continuous monitoring of population dynamics for both target and non-target species.

The arthropod community in maize ecosystems comprises distinct aboveground and subterranean components. Standard methodologies for assessing arthropod diversity typically incorporate multiple approaches, including visual census (direct observation), pitfall trapping, aerial pan trapping, yellow sticky card deployment, plant dissection, and soil core sampling [36,39,69]. The current investigation specifically employed visual census methodology to document the taxonomic composition and population dynamics of epigeal arthropod communities. It is important to acknowledge that the direct observation method employed in this field survey, despite providing essential firsthand data, is prone to certain inherent limitations. Firstly, the assessments of arthropod community could be influenced by subjective bias, both within and between observers. Secondly, the method’s demand for intensive labor restricted the spatial and temporal scope of our sampling, thereby risking an underrepresentation of transient or localized phenomena. Moreover, the field presence of the investigators might have inadvertently altered the natural environment, impacting the very behaviors and states being monitored. These factors collectively indicate that our results, though informative, must be considered within the context of these methodological constraints. In addition, it should be noted that this study design presents notable limitations: (1) the temporal constraint of a two-year monitoring period, (2) geographical restriction to a single experimental site, (3) only one investigation method was employed, and (4) differences in application of chemical pesticides and sampling site. Given that the ecological impacts of genetically modified crops manifest through gradual, cumulative processes spanning multiple generations, the methodological framework utilized here underscores the necessity for enhanced monitoring protocols. To establish scientifically robust conclusions regarding environmental safety assessments, future research initiatives should implement a multidimensional strategy featuring: (1) the integration of complementary ground-based and aerial sampling techniques, (2) long-term monitoring across extended temporal scales (>5 years), (3) multi-site comparative analyses encompassing diverse agro ecological regions, and (4) concurrent evaluation of Bt-receptor maize and Bt maize under standardized insecticide regimes.

## 5. Conclusions

This two-year field investigation (2023–2024) conducted in Bazhong City, Sichuan Province, utilized systematic field monitoring to compare arthropod community dynamics between non-Bt maize and Bt-Cry1Ab maize (DBN9936) cultivation systems. Field investigations revealed a notable absence of *M. cingulum*, a key larval parasitoid of *O. furnacalis*, in Bt maize plots compared to conventional counterparts. The populations of non-target herbivorous pests and natural enemies such as Aphididae, *C. sinica*, *F. tenuicornis*, and *O. sauteri* in Bt maize fields were higher than those in non-Bt maize fields, while the populations of target herbivorous pests such as *O. furnacalis* and *M. lorei* in Bt maize fields were lower than those in non-Bt maize fields. However, no significant differences (*p* > 0.05) were observed in ecological indices including the Simpson diversity index, Shannon–Wiener diversity index, Pielou evenness index, McIntosh diversity index, and community stability indices (*N*_n_/*N*_p_, *N*_d_/*N*_p_, and *S*_d_/*S*_p_). These findings demonstrate that Bt-Cry1Ab maize (DBN9936) cultivation effectively controls target pests without causing major alterations in overall community diversity, although specific taxa exhibited variable responses. This provides substantial empirical evidence to support its sustainable deployment in southern China’s agricultural landscapes.

## Figures and Tables

**Figure 1 insects-16-01132-f001:**
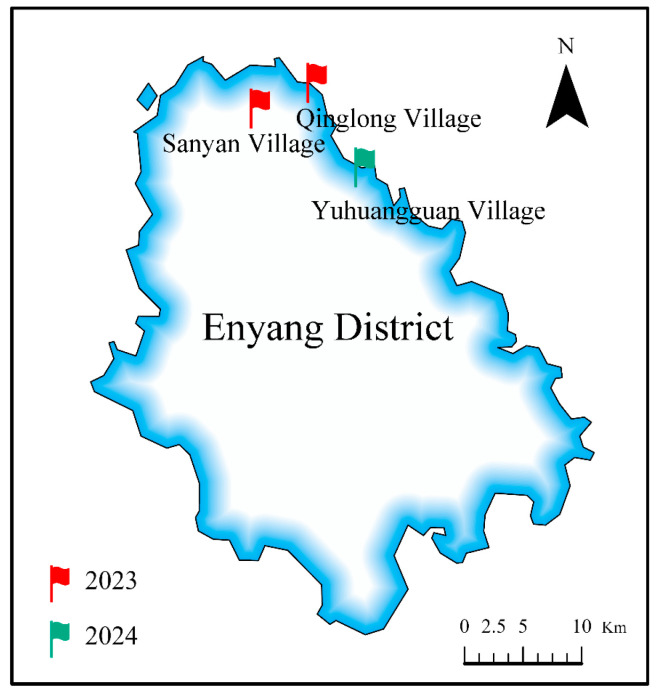
General characteristics of sampling sites for field investigation.

**Figure 2 insects-16-01132-f002:**
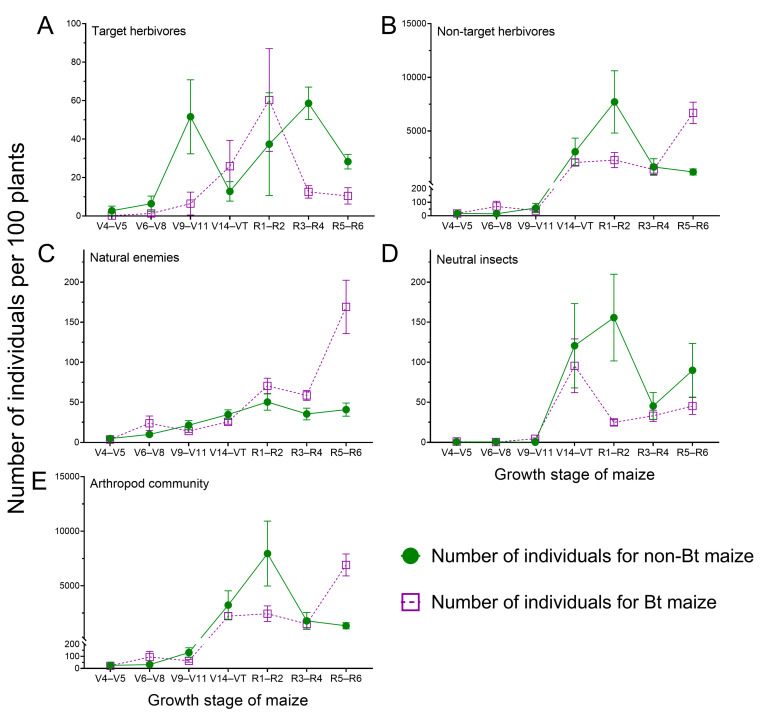
Number of individuals per 100 plants of four groups of arthropods in Bt and non-Bt maize fields. Data are presented as the mean ± standard error (SE).

**Figure 3 insects-16-01132-f003:**
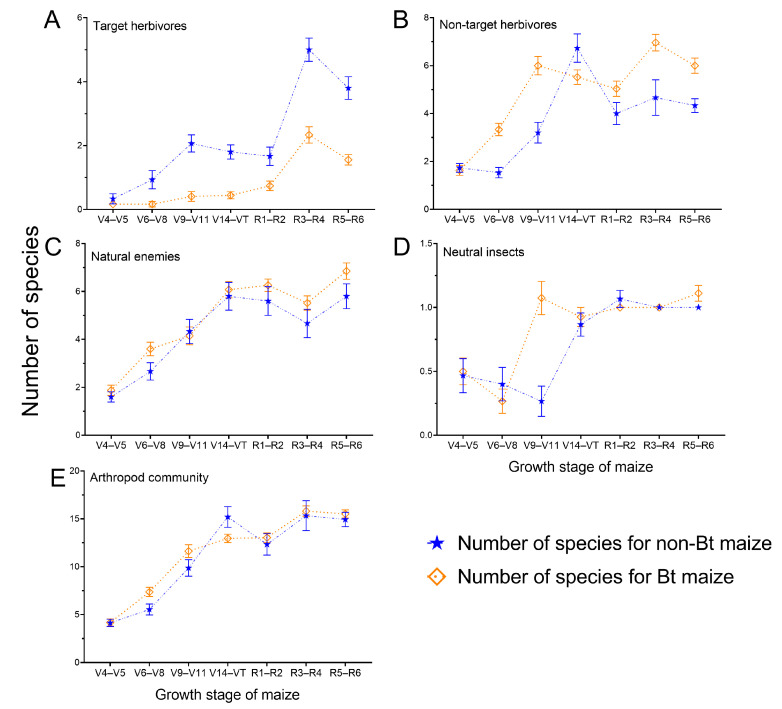
Number of species of four groups of arthropods in Bt and non-Bt maize fields. Data are presented as the mean ± standard error (SE).

**Figure 4 insects-16-01132-f004:**
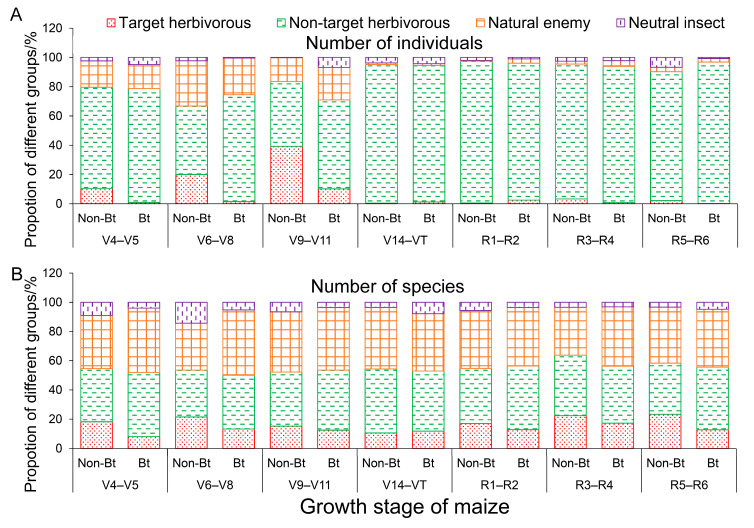
Proportional representation of the species number and individual number of different groups of arthropods found in Bt and non-Bt maize fields.

**Figure 5 insects-16-01132-f005:**
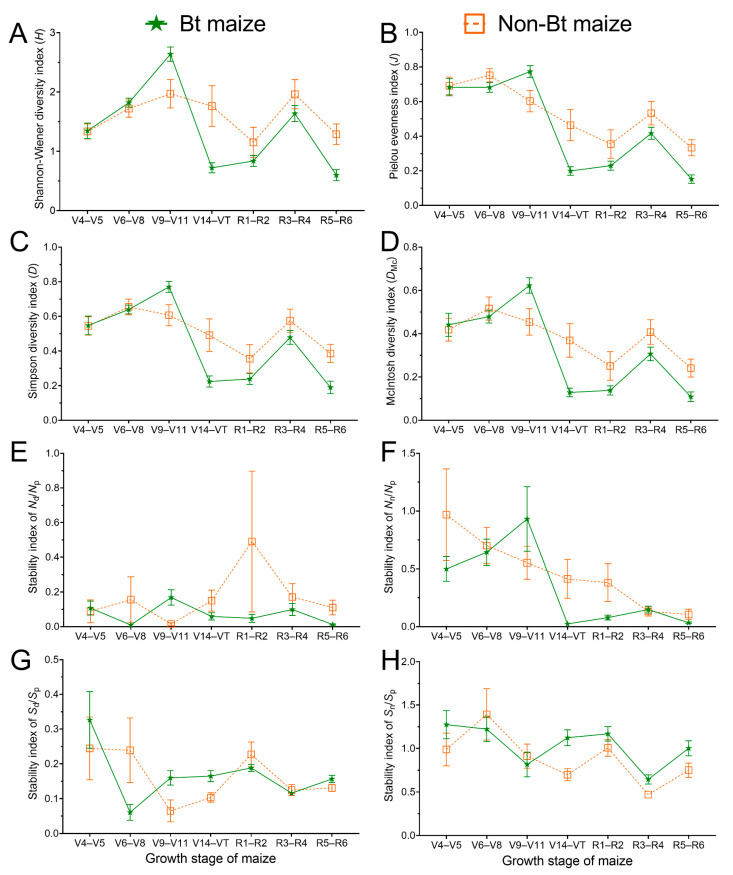
Diversity (**A**–**D**) and stability (**E**–**H**) indices of the arthropod community in Bt and non-Bt maize fields. Data are presented as the mean ± SE.

**Figure 6 insects-16-01132-f006:**
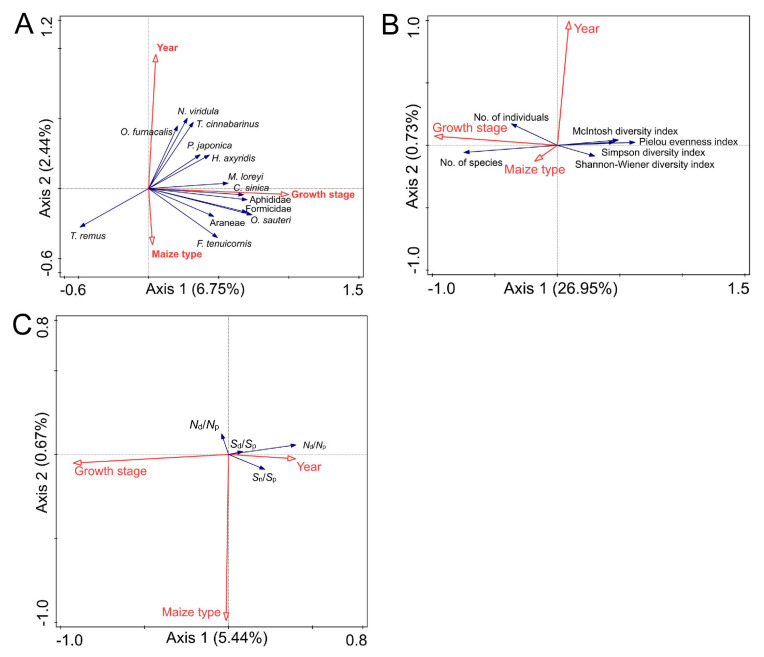
Redundancy analysis of ecological indices of the arthropod community and environmental factors. Abundance (**A**), diversity (**B**), and community (**C**) stability indices of the arthropod community. Each arrow points in the direction of the steepest increase in values for the environmental variable or ecological index of the arthropod community. The angle between arrows indicates the sign of the correlation between individual environmental variables and ecological indices: the approximated correlation is positive when the angle is sharp and negative when the angle is larger than 90°.

**Table 1 insects-16-01132-t001:** Sørensen index of the arthropod community in Bt and non-Bt maize fields.

Year	Growth Stage of Maize	Arthropod Community	Target Herbivores	Non-Target Herbivores	Natural Enemies	Neutral Insects
2023	V4–V5	0.44	0.50	0.46	0.31	1.00
	V6–V8	0.55	0.60	0.60	0.50	0.40
	V9–V11	0.75	0.77	0.67	0.84	0.67
	V14–VT	0.85	0.75	0.88	0.90	0.50
	R1–R2	0.83	0.89	0.80	0.87	0.67
	R3–R4	0.75	0.86	0.73	0.69	1.00
	R5–R6	0.84	0.86	0.75	0.93	0.67
2024	V4–V5	0.27	-	0.33	0.33	-
	V6–V8	0.55	-	0.67	0.67	-
	V9–V11	0.32	-	0.40	0.33	-
	V14–VT	0.67	0.50	0.50	0.80	1.00
	R1–R2	0.67	0.57	0.63	0.70	1.00
	R3–R4	0.77	0.80	0.63	0.84	1.00
	R5–R6	0.64	0.50	0.63	0.67	1.00

## Data Availability

The original contributions presented in this study are included in the article/Appendix A. Further inquiries can be directed to the corresponding author.

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
