# Peer review of "Effect of Bt-Cry1Ab Maize Commercialization on Arthropod Community Biodiversity in Southwest China"

_insects, 2025, doi:10.3390/insects16111132_

Round 1

Reviewer 1 Report

Comments and Suggestions for Authors

The MS used a two-year experiment to investigate the composition of the arthropod community in the aboveground parts of Bt-Cry1Ab maize (DBN9936) and its control plots. The surveyed groups included both target and non-target arthropods. Through significant difference analysis, similarity analysis, stability analysis, and RDA analysis, the impact of Bt-Cry1Ab maize (DBN9936) on the community biodiversity of arthropods was comprehensively analyzed. It showed that Bt Cry1Ab maize (DBN9936) cultivation maintains functional arthropod community structure while effectively controlling target pests. It will provide important information for the safety of Bt maize in Southwest China. The comments are as follows:
Q1.Intrduction: In my opinion, the main point I got from the MS was that the effect of genetically modified corn cultivation on the biodiversity of above-ground arthropods has been studied. However, in the introduction section, there is relatively little information on the research progress in this area both domestically and internationally. It is suggested to add relevant introductions to this part.
Q2. Line273, Table 1 and Line 355 Figure 2,The "No." is an abbreviation and its meaning should be explained.

Q3. Line 322, ”The of target herbivorous insect species 322 were significantly lower than that in non-Bt maize fields during V6–V8”should be “The number of target herbivorous insect species 322 were significantly lower than that in non-Bt maize fields during V6–V8”.

Q4. Line 355, In this paragraph, "Similarity coefficient" refers to "Sørensen index" as mentioned in line 226? Please maintain the consistent expression.

Q5.Line 419-420 and Line518-522,Excessive description of the research background. It is suggested that the amount can be appropriately reduced.

Author Response

Comments and Suggestions for Authors

The MS used a two-year experiment to investigate the composition of the arthropod community in the aboveground parts of Bt-Cry1Ab maize (DBN9936) and its control plots. The surveyed groups included both target and non-target arthropods. Through significant difference analysis, similarity analysis, stability analysis, and RDA analysis, the impact of Bt-Cry1Ab maize (DBN9936) on the community biodiversity of arthropods was comprehensively analyzed. It showed that Bt Cry1Ab maize (DBN9936) cultivation maintains functional arthropod community structure while effectively controlling target pests. It will provide important information for the safety of Bt maize in Southwest China. The comments are as follows:

Q1. Intrduction: In my opinion, the main point I got from the MS was that the effect of genetically modified corn cultivation on the biodiversity of above-ground arthropods has been studied. However, in the introduction section, there is relatively little information on the research progress in this area both domestically and internationally. It is suggested to add relevant introductions to this part.

Response: Revised as suggested, as such: 'Over the past 25 years, hundreds of environmental impact studies on genetically modified crops conducted across Europe and the Americas have indicated that Bt corn exerts minimal, predominantly neutral effects on non-target invertebrate com-munities in cornfields [7, 33, 34]. A key finding was that robust and stable food webs persisted across diverse Bt maize fields, mirroring the patterns observed in conventional maize fields [35]. A significant lacuna exists, however, in the data from Asian agricultural systems. Empirical evidence from confined field trials in China has indicated that Bt maize cultivars exert no significant adverse impacts on arthropod diversity at experimental scales [19, 36-39]. A recent study demonstrates that Bt maize has no adverse effects on predatory natural enemy diversity within the Yellow-Huai-Hai summer maize-growing region of China [40].' (Lines 132-141)

Q2. Line273, Table 1 and Line 355 Figure 2, The "No." is an abbreviation and its meaning should be explained.

Response: Revised as suggested. The abbreviation 'No.' has been revised to 'Number' in the figures. Additionally, Table 1 is now presented as Table S1, with an expanded caption specifying that 'No.' stands for 'the number of individuals'.

Q3. Line 322, The of target herbivorous insect species were significantly lower than that in non-Bt maize fields during V6–V8” should be “The number of target herbivorous insect species were significantly lower than that in non-Bt maize fields during V6–V8”.

Response: Revised as suggested.

Q4. Line 355, In this paragraph, "Similarity coefficient" refers to "Sørensen index" as mentioned in line 226? Please maintain the consistent expression.

Response: Revised as suggested. The 'Similarity coefficient' has been standardized to 'Sørensen index' throughout the manuscript.

Q5. Line 419-420 and Line 518-522, Excessive description of the research background. It is suggested that the amount can be appropriately reduced.

Response: Revised as suggested. We have now removed ' The cultivation of genetically modified crops, particularly Bt varieties, has demonstrated significant economic benefits in agricultural production through reduced chemical pesticide use and the enhanced protection of beneficial insects [11, 20] ' and 'This technology not only effectively manages target pests like O. furnacalis and diminishes the reliance on chemical pesticides but also reduces the incidence of bacterial rot in maize, enhances crop yield and quality, conserves beneficial insect populations.'

Reviewer 2 Report

Comments and Suggestions for Authors

The present manuscript explores the ecological effects of Bt-Cry1Ab maize (DBN9936) on field arthoropod community biodiversity in Sichuan Province, China. The study addresses a relevant question regarding the environmental safety of newly commercialized transgenic maize under real-world cultivation. The large sampling effort represents a valuable dataset that could contribute to risk assessment frameworks for Bt crops in Asia. However, despite the extensive field observations, the experimental design, statistical analysis , and interpretive balance could be improved before considering the paper for publication.

Major comments

1) Experimental design:

  • Spatial pseudorreplication (different villages) is a limitation, so this effect should be stated, also non-Bt maize as control is not near-isogenic, so this genetic background could be influencing the results. Should also be stated or discussed in discussion to improve the manuscript.
  • the use of insecticides might reflect pesticide effects more than Bt-crop effects, so should be acknowledged as a major limitation in interpreting ecological differences.

2) Statistical analysis:

  • Authors used multifactorial ANOVA, although GLM or GLMMs would be more appropiate.

3) Figure and tables. Please revise Table 1, is too dense and redundant, consider summarizing to key taxa

4) Results. Concluding that Bt maize "maintains functional arthropod community structure" might be too strong given the results shown here.Alternative for discussion: "no major alterations in overall diversity indices were detected, although specific taxa showed variable responses"

Author Response

The present manuscript explores the ecological effects of Bt-Cry1Ab maize (DBN9936) on field arthoropod community biodiversity in Sichuan Province, China. The study addresses a relevant question regarding the environmental safety of newly commercialized transgenic maize under real-world cultivation. The large sampling effort represents a valuable dataset that could contribute to risk assessment frameworks for Bt crops in Asia. However, despite the extensive field observations, the experimental design, statistical analysis, and interpretive balance could be improved before considering the paper for publication.

Response: We are thankful to you for a positive assessment of our submission and critical points to improve the work for readers. We have accepted all your suggestions, and revised all these contents in this revised manuscript. Please see the revised manuscript and all the modification contents have highlighted using the track change mode in MS word.

Major comments

1) Experimental design:

  • Spatial pseudorreplication (different villages) is a limitation, so this effect should be stated, also non-Bt maize as control is not near-isogenic, so this genetic background could be influencing the results. Should also be stated or discussed in discussion to improve the manuscript.

Response: Revised as suggested. We have emphasized in materials and methods, and discussion that the primary objective of this study was to evaluate the impact of Bt maize and non-Bt maize production patterns on arthropod community biodiversity, while also acknowledging the inherent limitations of our experimental design. As such:

 'The study utilized the Bt-Cry1Ab maize Zhongdan 808D (event DBN9936, Bt-maize), an early-adopted and widely promoted cultivar in Sichuan Province, China. Seed for this variety was provided by DBN Group (Beijing, China) in 2023 and by Bei-jing Abundance High-tech Seed Industry Co., Ltd. (Beijing, China) in 2024. For com-parison, two non-Bt maize varieties (Dongdan 100 and Shuyu 336) conventionally planted by local farmers were included, sourced from Liaoning Dongya Seed Industry Co., Ltd. (Shenyang, China) and Sichuan Shuyu Technology Agriculture Development Co., Ltd. (Chengdu, China), respectively.' (Lines 156-163)

 'To address this knowledge gap, this study comprehensively evaluated the effects of Bt maize and non-Bt maize production systems on arthropod community biodiversity in Sichuan Province, a representative southern maize production region characterized by a subtropical monsoon climate and a mosaic agricultural landscapes.' (Lines 483-487)

 'In addition, it should be noted that this study design presents three notable limitations: (1) the temporal constraint of a two-year monitoring period, (2) geographical restriction to a single experimental site, and (3) only one investigation method was employed, and (4) differences in application of chemical pesticides and sampling site. Given that the ecological impacts of genetically modified crops manifest through gradual, cumulative processes spanning multiple generations, the methodological framework utilized here underscores the necessity for enhanced monitoring protocols. To establish scientifically robust conclusions regarding environmental safety assessments, future research initiatives should implement a multidimensional strategy featuring: (1) the integration of complementary ground-based and aerial sampling techniques, (2) long-term monitoring across extended temporal scales (>5 years), (3) multi-site comparative analyses encompassing diverse agro ecological regions, and (4) concurrent evaluation of Bt-receptor maize and Bt maize under standardized insecticide regimes.'(Lines 604-616)

  • the use of insecticides might reflect pesticide effects more than Bt-crop effects, so should be acknowledged as a major limitation in interpreting ecological differences.

Response: Revised as suggested. We have emphasized the inherent limitations of the use of insecticides in discussion, as such: 'Our results also showed a higher population density of non-target insects in Bt maize fields compared to non-Bt maize fields. The abundance of target herbivores, non-target herbivores, natural enemies, neutral insects and arthropods varied significantly among the interaction effects of year × growth stage, year × maize type, or growth stage × maize type. A plausible explanation for the observed patterns involves the application of the lepidopteran-specific insecticide chlorantraniliprole in non-Bt maize fields, which was absent in Bt plots. This insecticide may have broadly sup-pressed a range of non-target insects beyond the intended lepidopteran pests. In addition, geographical variation across sampling sites and differences in replication numbers may have contributed to the ecological differences detected between the two cropping systems.' (Lines 576-585)

2) Statistical analysis:

  • Authors used multifactorial ANOVA, although GLM or GLMMs would be more appropiate.

Response: Revised as suggested. We have now switched from multifactorial ANOVA to the Generalized Linear Model (GLM). Please see the revised manuscript and all the modification contents have highlighted by using the track changes mode in MS word. (Lines 278-375)

3) Figure and tables. Please revise Table 1, is too dense and redundant, consider summarizing to key taxa

Response: Table 1 is now presented as Table S1.

4) Results. Concluding that Bt maize "maintains functional arthropod community structure" might be too strong given the results shown here. Alternative for discussion: "no major alterations in overall diversity indices were detected, although specific taxa showed variable responses"

Response: Revised as suggested. We have now changed this sentence to 'These findings demonstrate that Bt-Cry1Ab maize (DBN9936) cultivation effectively controls target pests without causing major alterations in overall community diversity, although specific taxa exhibited variable responses. This provides substantial empirical evidence to support its sustainable deployment in southern China's agricultural landscapes' (Lines 629-633)

Round 2

Reviewer 2 Report

Comments and Suggestions for Authors

The authors have addressed all my comments and suggestions, so I propose accepting the manuscript in its current form.